genetics/evolution

Pacific prehistory, process models, migration, dominance

**Authors for correspondence:**
J. Stephen Lansing
e-mail: lansing@santafe.edu
Murray P. Cox
e-mail: m.p.cox@massey.ac.nz

†These authors contributed equally to the study.

# Sex-linked genetic diversity originates from persistent sociocultural processes at microgeographic scales

Ning Ning Chung[1,3,†], Guy S. Jacobs[1,†], Herawati Sudoyo[4,5,6], Safarina G. Malik[4], Lock Yue Chew[1,2], J. Stephen Lansing[7,8] and Murray P. Cox[9,10]

[1]Complexity Institute, and [2]School of Physical and Mathematical Sciences, Nanyang Technological University, Singapore
[3]Centre for University Core, Singapore University of Social Sciences, Singapore
[4]Genome Diversity and Diseases Laboratory, Eijkman Institute for Molecular Biology, Jakarta, Indonesia
[5]Department of Medical Biology, Faculty of Medicine, University of Indonesia, Jakarta, Indonesia
[6]Sydney Medical School, University of Sydney, Sydney, New South Wales 2006, Australia
[7]Santa Fe Institute, Santa Fe, NM 87501, USA
[8]Stockholm Resilience Center, Kräftriket, 10405 Stockholm, Sweden
[9]Statistics and Bioinformatics Group, School of Fundamental Sciences, Massey University, Palmerston North 4410, New Zealand
[10]Te Pūnaha Matatini, Centre of Research Excellence for Complex Systems, Aukland, New Zealand

MPC, 0000-0003-1936-0236

Population genetics has been successful at identifying the relationships between human groups and their interconnected histories. However, the link between genetic demography inferred at large scales and the individual human behaviours that ultimately generate that demography is not always clear. While anthropological and historical context are routinely presented as adjuncts in population genetic studies to help describe the past, determining how underlying patterns of human sociocultural behaviour impact genetics still remains challenging. Here, we analyse patterns of genetic variation in village-scale samples from two islands in eastern Indonesia, patrilocal Sumba and a matrilocal region of Timor. Adopting a 'process modelling' approach, we iteratively explore combinations of structurally different models as a thinking tool. We find interconnected socio-genetic interactions involving sex-biased migration, lineage-focused founder effects, and on Sumba, heritable social dominance. Strikingly, founder ideology, a cultural model derived from anthropological and

archaeological studies at larger regional scales, has both its origins and impact at the scale of villages. Process modelling lets us explore these complex interactions, first by circumventing the complexity of formal inference when studying large datasets with many interacting parts, and then by explicitly testing complex anthropological hypotheses about sociocultural behaviour from a more familiar population genetic standpoint.

# 1. Introduction

Over the last decade, genetics has transformed our knowledge of the last great human migration into pristine environments—the settlement of the greater Pacific region [1,2]. However, by focusing on the timing and geography of migration routes, these studies summarize the outcome of historical processes, but do not explain them. All large-scale genetic patterns ultimately originate at the level of social interactions within and between communities, the domain of archaeologists and anthropologists. In principle, genetics should also be useful for analysing patterns at this scale [3–6], but so far, there have been few attempts to use genetic data to evaluate competing hypotheses for the origins of movements [7–9], and more generally, of social change [10,11]. To do so requires a shift from descriptive to process-driven modelling, which uses genetic information to directly test anthropological hypotheses.

Sex-specific processes can create different patterns of genetic diversity at sex-linked loci [12–16] with an extensive literature reporting such patterns in humans (e.g. [17–23]). Recent work is providing increasingly precise descriptions of the genetic impact of kinship practices [24], as well as the genetic relationships between communities at fine geographical scales [25], while advances in modelling are revealing both the drivers of changes in kinship practices at global scales [26,27] and surprising diversity in the rate and manner of change more locally [28]. Despite these advances, discussion of social processes impacting genetic diversity commonly focuses on the proximate genetic cause—sex-biased migration—with little emphasis on the sociocultural processes ultimately driving this behaviour. A growing body of the literature shows that sociocultural actions, extending beyond sex-biased migration to the formation of patrilineal kin groups and competition among individuals for status, can create differential patterns of sex-linked diversity at global [29,30] and regional scales [31,32]. What is less clear is how these patterns arise at very small geographical scales, and how the details of local social behaviour cascade up to more easily observed global genetic patterns.

The greater Pacific region, rich in sociocultural complexity from the Pleistocene [33,34] to the present [35,36], is an ideal empirical setting to explore sex-linked genetic diversification at extremely small geographical scales [2,37]. We take as our case study the neighbouring islands of Sumba and Timor, which have already yielded insights into the role of microevolution in genetic and cultural diversity [11,38]. Within each island, most villages are located only a few kilometres apart, but astonishingly, village populations are still distinguishable genomically [25]. Although all of these villages ultimately derive from the matrilocal communities of the Austronesian farming expansion [39], the two islands are characterized today by their contrasting post-marital residence rules [28]: patrilocal on Sumba and matrilocal in the Wehali region of Timor. Otherwise, both islands have similar genetic histories, environmental settings, cultural backgrounds and historical trajectories [2,5,40,41]. Here, we evaluate the power of anthropological models to predict diversity at sex-linked loci in 23 villages located on Sumba and Timor. We demonstrate a remarkable parallel in the sociocultural behaviours that are involved in generating diversity at the micro-scale with higher level anthropological and archaeological models of Pacific colonization.

# 2. Material and methods

## 2.1. Data

The genetic data analysed in this study consist of mitochondrial DNA (mtDNA) sequences [11,21,22] (positions 16 001–16 540 of the revised Cambridge Reference Sequence [42]) and haplogroup-informative SNPs screened hierarchically on the Y chromosome together with 12 Y-STRs [21,22,40,43]. These haploid markers were chosen purposely as they record sex-specific processes, and in contrast to nuclear markers, they vary over the short time frames considered in this study. Only men were sampled because they carry both mtDNA and a Y chromosome. Samples originated from 14 villages

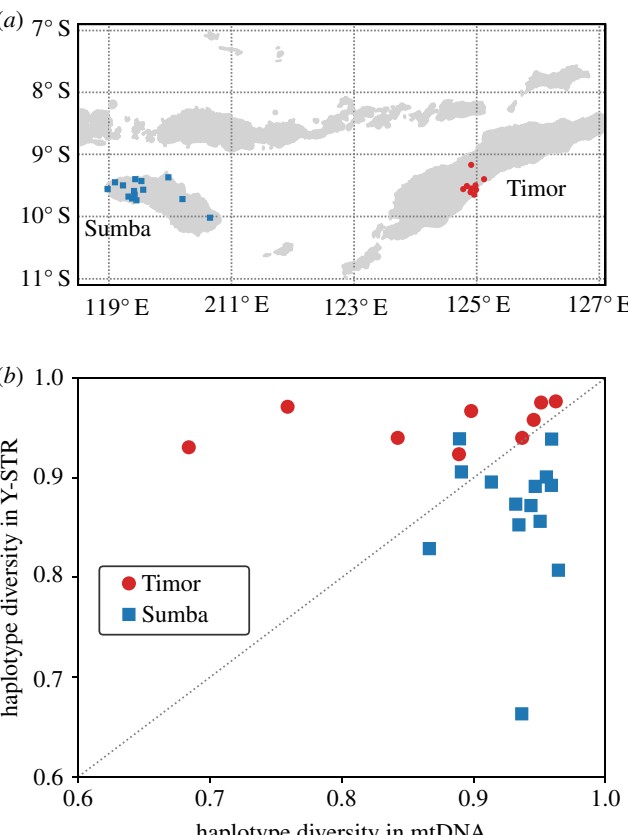

**Figure 1.** (*a*) Map of sampling locations on Sumba and Timor in eastern Indonesia. (*b*) Levels of sample haplotype diversity on mtDNA and the Y chromosome are inversely related in the patrilocal communities of Sumba (blue squares) and the matrilocal communities of Timor (red circles). Note that the two datasets are nearly symmetrical. When two datasets are reflection-symmetric with respect to a common axis, then the summation of the mean of their projections onto the normal line of the axis is zero. Here, $\langle D_S \rangle + \langle D_T \rangle = -0.02$.

on Sumba (634 mtDNA, 646 Y chromosome), and nine villages on neighbouring Timor (450 mtDNA, 421 Y chromosome) (figure 1*a*).

## 2.2. Model fitting and statistical analyses

Our analysis characterizes the data using two statistics that (i) capture essential properties of genetic variation within the sample and (ii) are able to differentiate between the sociocultural models that we explore. We measured the sample haplotype diversity (often called 'heterozygosity' for diploid loci) to assess the combined effects of mutation, migration and drift, and Slatkin's [44] exact test to characterize the extent to which the haplotype frequency spectrum deviates from neutral expectations. Sample haplotype diversity is calculated as usual

$$H = 1 - \sum_{i=1}^{k} p_i^2, \tag{2.1}$$

where $k$ is the number of haplotypes and $p_i$ is the frequency of each respective haplotype.

Slatkin's exact test [44] is based on Ewens' sampling formula [45] and can detect heritable reproductive skew [46]. Specifically, communities experiencing reproductive skews tend to have haplotype frequency distributions that, over time, become unlikely under the neutral model, with an excess of common haplotypes in patrilines or matrilines, depending on whether the community experiences male or female dominance. Such skewed distributions provide evidence for cultural selection, defined as the heritable transmission of any kind of behaviour that affects reproductive success [47]. We calculated Slatkin's test using the Monte Carlo function with $10^6$ samples and the code provided at http://ib.berkeley.edu/labs/slatkin/monty/Ewens_exact.program [44].

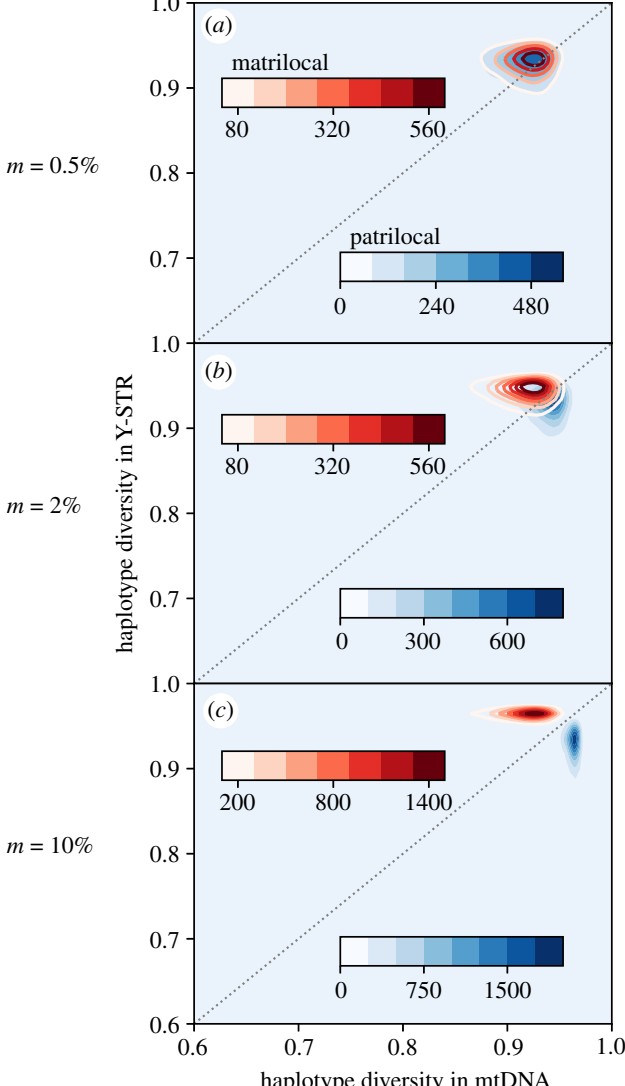

**Figure 2.** Effects of post-marital residence movements on sample haplotype diversity. (*a*) The migration rate is 0.5% per generation for men in matrilocal villages and women in patrilocal villages, with a low rate of 0.5% for the opposite sex to accommodate low levels of non-compliance with the marriage rules. (*b,c*) Show the effects of increasing the migration rate to 2% and 10%, with the non-compliant migration rate held steady. The contour lines are estimated based on $10^4$ data points simulated under the island model. Data points are located on a $100 \times 100$ uniform grid. The value of the contour line at a given location $(x_c, y_c)$, after being divided by the number of simulations ($10^4$), thus gives the probability that a data point is found within the grid location ($x_c - 0.0025 \leq x \leq x_c + 0.0025$, $y_c - 0.0025 \leq y \leq y_c + 0.0025$). The rest of the parameters used are $N = 300$, $V = 50$, $\mu_{mt} = 0.0186$, $\mu_Y = 0.0249$, $T = 1000$ and $n_s = 40$.

Simulations confirm that haplotype diversity patterns clearly distinguish sex-biased migration (figures 2–4), while Slatkin's test can accurately identify the signal of social dominance (figure 5), such that together these statistics offer powerful insight into sociocultural practices.

To refine our understanding of diversity at the village scale, we explored these statistics within increasingly complex sociocultural scenarios modelled using forward-time simulation implemented in Python. The three sociocultural features of our model are sex-biased migration, social dominance and serial founder effects. Model specifications and parameters are given in the text, and the model code is available at https://github.com/nnchung/Migration-Models/releases/tag/v1.0.

To assess the performance of the different sociocultural models, we use a multidimensional distance metric based on the unidimensional Kolmogorov–Smirnov (KS) test. This metric describes the difference between a set of data (the observed distributions of summary statistics) and a set of given probability distributions (the model predictions). For two-dimensional data, the KS statistic is the maximum difference between the quadrant-integrated probabilities of the data and the model [48]. Specifically,

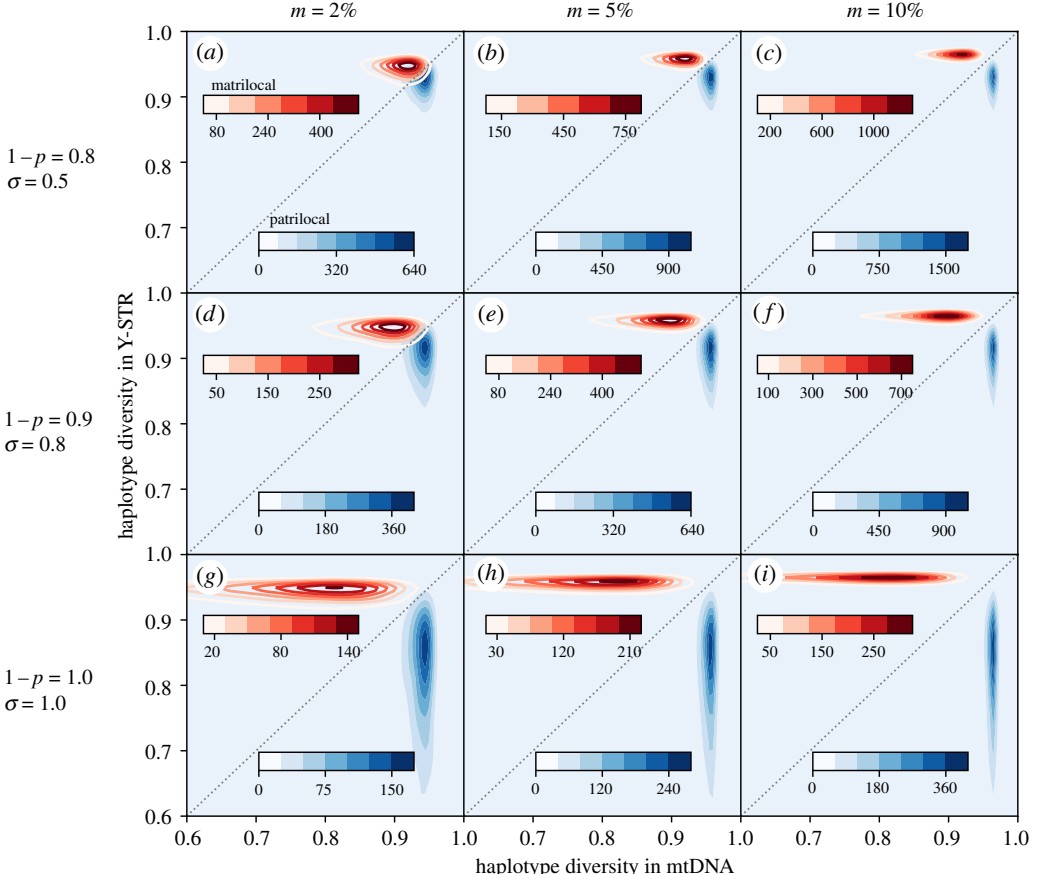

**Figure 3.** Effects of adding dominance to the migration model. The migration rate of the dispersing sex is set to 2%, 5% and 10% for the first, second and third columns, with a constant low level of movements not conforming with the marriage rules (0.5%). The rows simulate populations with weak ($1 - p = 0.8$, $\sigma = 0.5$), intermediate ($1 - p = 0.9$, $\sigma = 0.8$) and strong ($1 - p = 1.0$, $\sigma = 1.0$) dominance. The rest of the parameters are $N = 300$, $\mu_{mt} = 0.0186$, $\mu_Y = 0.0249$, $n_s = 40$ and $\delta = 0.06$. The contour lines are estimated based on $10^4$ data points simulated using the island dominance model.

the two-dimensional space where the data lie is divided into four quadrants around a point $(x_i, y_i)$. The four quadrants are: (1) $(x > x_i, y > y_i)$, (2) $(x < x_i, y > y_i)$, (3) $(x > x_i, y < y_i)$ and (4) $(x < x_i, y < y_i)$. The integrated probability is then calculated within each quadrant for the data and model, and differences between the integrated probabilities of the data and model are calculated for the four quadrants. This process is repeated for 225 (=15 × 15) uniformly distributed points in the state space. We then find the point $(x_c, y_c)$ for which the four quadrants around it maximize the difference between the quadrant-integrated probabilities of the data and the model. The maximum difference gives the two-dimensional KS statistic for the model.

# 3. Results

## 3.1. Diversity patterns within Sumba and Timor

We study mtDNA and the Y chromosome in two matched sets of communities: 14 patrilocal villages on Sumba, where newly-wed couples move to the community of the husband, and nine matrilocal villages on Timor, where newly-wed couples move to the community of the wife (figure 1a). This difference in sex dispersal rules leaves contrasting patterns of diversity, particularly revealed in the sample haplotype diversity of mtDNA and the Y chromosome (figure 1b). Each point on the plot shows the genetic diversity of individuals within a single village; blue squares for patrilineal Sumba, and red circles for the matrilineal region of Timor. The axes show the level of haplotype diversity (variation) observed on the mtDNA, inherited from the mother, and on the Y chromosome, inherited from the father. In patrilocal Sumba, men tend to remain in their natal village and seek wives from further afield. This

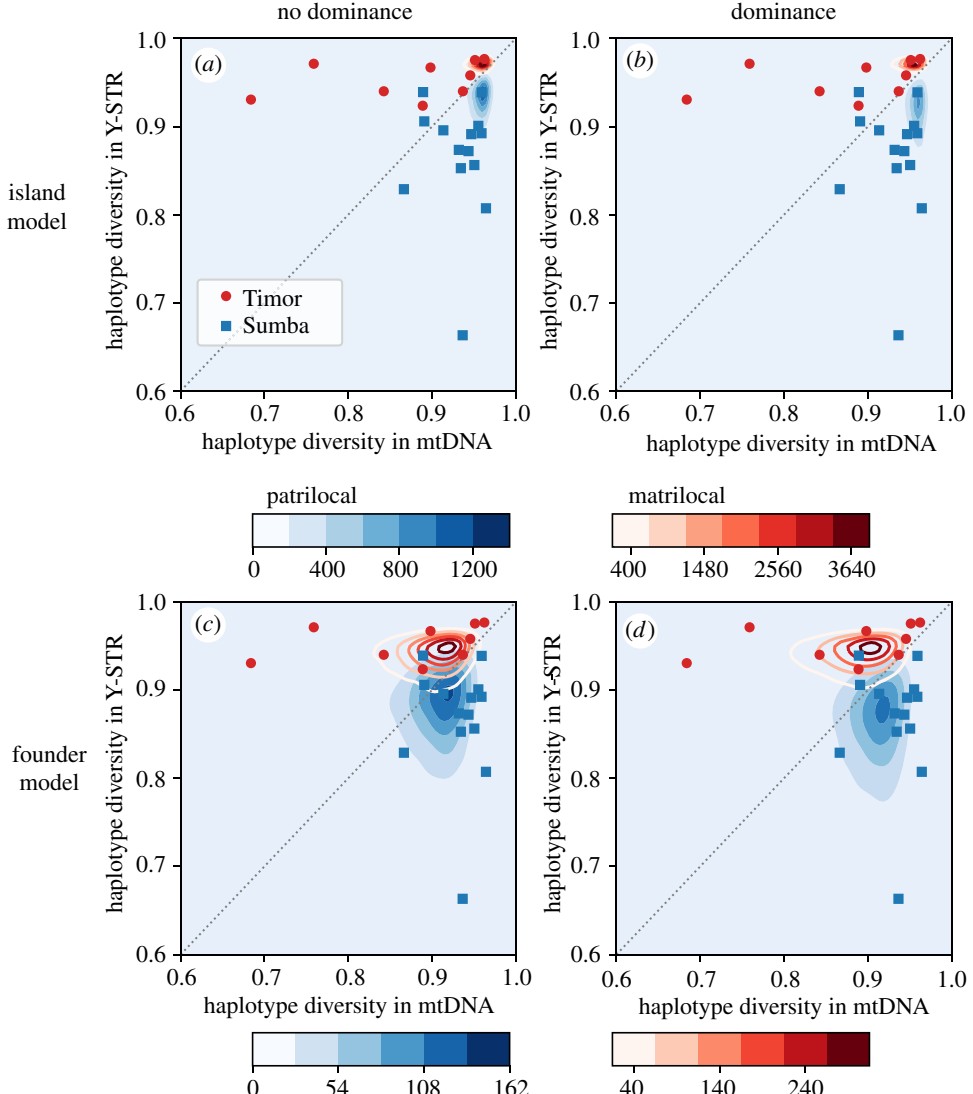

**Figure 4.** (a,b) Haplotype diversity for the island model and (c,d) the founder model. Plots on the left are without dominance, while those on the right include the effects of dominance. Parameters used are shown in table 1.

results in a bias: there is more genetic variation among the women, who move more, than the men, who stay localized. The reverse is true in matrilineal Timor: there is greater haplotype diversity among the husbands and fathers, who move into the village.

Differential diversity at male- and female-linked genetic loci is commonly reported in the literature [14,15,30]. The new observation here is that the sex-biased skew on male and female inherited loci is a mirror-image for the nearby communities of matrilocal and patrilocal villages. In previous analyses of a different measure of diversity (the mismatch distribution), we observed that the tendency for men to stay in their natal village on marriage is greater on patrilocal Sumba than the tendency for women to stay put on matrilocal Timor [38]. Given this, why are the skewed distributions of villages from the two islands so variable within each marriage class, yet also so symmetrical between them? Could some other sociocultural processes—perhaps social dominance or founder history, as predicted by Bellwood *et al.* [49]—be impacting diversity patterns?

## 3.2. Theoretical diversity given sex-biased migration

To answer these questions, we began by exploring the theoretical implications of sex-biased migration alone. We first implemented an island model [50], in which the total population is assumed to be divided into $V$ subgroups of size $N$ with random mating within subgroups following the standard Wright–Fisher model and migration between subgroups at a constant rate $m$. We modelled the

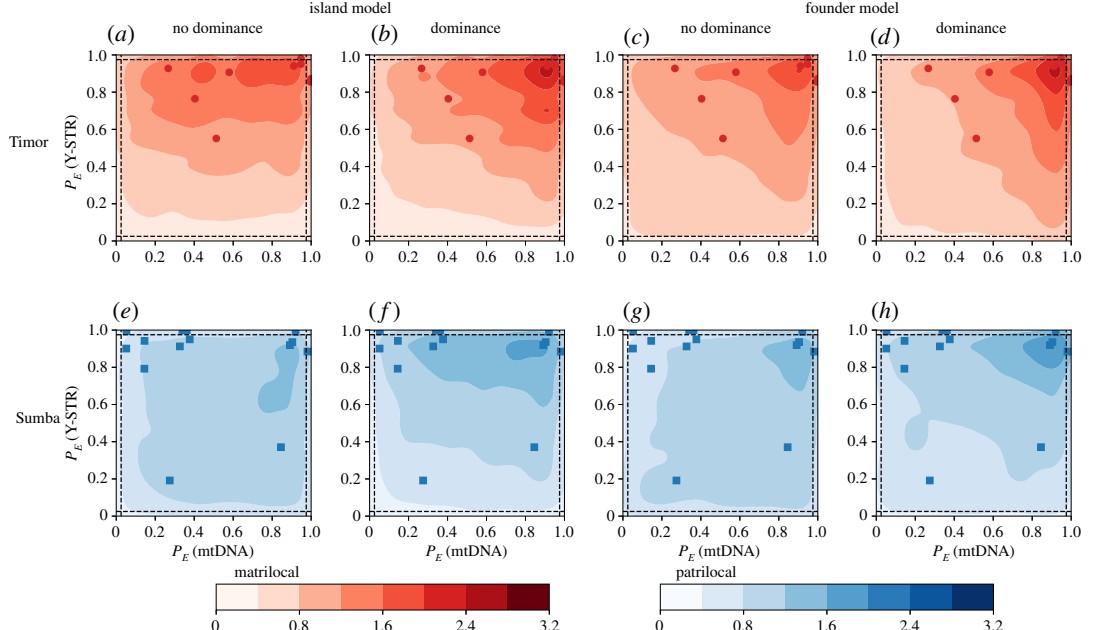

**Figure 5.** Results of Slatkin's test ($P_E$ values) based on mitochondrial (*x*-axis) and Y-STR (*y*-axis) variation, for individual villages (plotted points, red circles—matrilocal Timor, blue squares—patrilocal Sumba) and calculated on simulated data (red or blue shading, with more intense shading corresponding to a greater density of simulated villages). Dominance and founder effects are expected to generate haplotype frequency spectra with high $P_E$ values, and thus generate more pronounced skews in the $P_E$ value distribution.

populations of maternally inherited (mtDNA) and paternally inherited (Y chromosome) separately, with different migration and mutation rates. Mutation of haplotypes follows an infinite alleles model, such that each mutation gives rise to a novel haplotype. Mutation rates were set to 0.0186 per generation for the mtDNA hypervariable region (estimated from pedigrees [51]) and 0.0249 for the Y chromosome, based on the 12 STRs examined in Lansing *et al.* [52]. At the start of a simulation run, each individual has a distinct haplotype; the simulation is then run for 1000 generations, which is sufficiently long to reach an equilibrium state.

We explored a range of migration rates in a simulation of 50 villages of population size 300, taking a sample of 40 individuals from 14 patrilocal and nine matrilocal villages. These demographic parameters are close to the real averages for both Timor and Sumba [52]. If the marriage rules are adhered to strictly, women in matrilocal villages and men in patrilocal villages never move. Here, we treat the marriage rules with sufficient flexibility to permit a low rate of 0.5% migration per generation for women in matrilocal villages and men in patrilocal villages. By contrast, the migration rate for men in matrilocal villages and women in patrilocal villages is set between 0.5% and 10% per generation to study the effects of sex-biased migration.

Figure 2*a* simulates migration (husbands into matrilocal villages or brides into patrilocal villages) at a low rate of 0.5% per generation. As expected at these low rates of movement, no differences in diversity between men and women are seen.

Figure 2*b* simulates migration at 2% per generation. Now there is a slight bias in genetic variation among women versus men in the patrilocal villages, where the mean level of haplotype diversity of mtDNA (0.936) is very slightly higher than that of the Y chromosome (0.926). A similar level of bias (Y chromosome 0.942, mtDNA 0.912) is observed among men versus women in the matrilocal villages.

Increasing the migration rate to 10% results in a significant bias in the variation of mtDNA versus the Y chromosome in both patrilocal and matrilocal villages (figure 2*c*). The mean level of haplotype diversity of mtDNA in the patrilocal villages increases to 0.962, while haplotype diversity on the Y chromosome in the matrilocal villages increases to 0.963.

The overall picture is that higher migration *m* of the dispersing sex leads to a more biased pattern, while simultaneously causing a reduction in the overall variance of haplotype diversity (cf. figure 2*c* versus 2*a*). These differences suggest that a symmetric haplotype diversity pattern is unexpected given different degrees of sex-biased migration on Sumba and Timor, when different Y chromosome and mtDNA mutation rates are taken into account. Furthermore, simulated estimates of haplotype

diversity are all substantially higher than the observed average village haplotype diversity in matrilocal Timor (non-dispersing mtDNA $\bar{H}_{mtDNA} = 0.874$ versus $\bar{H}_Y = 0.954$) and patrilocal Sumba (non-dispersing Y $\bar{H}_Y = 0.866$ versus $\bar{H}_{mtDNA} = 0.932$). Even with other values of $m$, simulations are unable to generate villages with the extremely low haplotype diversity observed in the data (e.g. on Timor, the village of Kakaniuk has mtDNA haplotype diversity of 0.684, while on Sumba, the village of Kodi has Y chromosome haplotype diversity of 0.663). Reducing haplotype diversity in this simulation, while keeping the mutation rates constant, would require much lower migration rates, much smaller village population sizes or many fewer villages. As these demographic parameters are not realistic, the implication is that the island model with sex-biased dispersal alone is an incomplete description of the processes driving sex-linked genetic diversity on Sumba and Timor.

## 3.3. Potential sociocultural causes of low haplotype diversity

That a simple model of post-marital movement does not capture the diversity of village haplotype diversity observed in the data is perhaps surprising. The colonization of an individual island might be expected to have proceeded rapidly as populations spread into initially unoccupied or sparsely populated territory, such that all villages would have a similar—and long—time to recover haplotype diversity after the initial founder event. Any further immigration from outside the island, potentially impacting some villages but not others, would act to raise, rather than lower, genetic diversity. This is the opposite of the pattern we observe.

However, the anthropology of the region offers hints about the processes likely to be responsible for this low diversity. Two processes described by anthropologists offer alternative models. First, social dominance—a cultural Darwinian process in which some individuals become socially dominant and produce more surviving children than most other people in their village [53]. Second, archaeologist Peter Bellwood's 'founder-focused ideology' [49], where junior members of dominant lineages collect followers (again, often younger sons) and venture forth to found new communities. If these social processes were strong and persisted through time, either could potentially cause the observed skew in diversity.

To illustrate the potential effects of sociocultural factors, we began by building a combined model of dominance and sex-biased migration, and explored its theoretical behaviour. We followed the model of reproductive skew caused by dominance described in Lansing et al. [52], in which some subset of the village $\delta$ is socially dominant and more likely to reproduce. In the Wright–Fisher model, this corresponds to non-uniform probabilities of individuals being chosen as fathers. The probability that a dominant individual is chosen as a father is

$$\frac{1+\sigma}{N(1+\sigma\delta)} \tag{3.1}$$

while that for a non-dominant individual is

$$\frac{1}{N(1+\sigma\delta)} \tag{3.2}$$

where $\sigma$ is the selective advantage of the individual, and $\delta$ is the fraction of the population that has a selective advantage over the non-dominant individuals. As noted in Lansing et al. [52], membership of reproductively dominant groups is ultimately transient: men, therefore, inherit dominance from their fathers with probability $1-p$ in patrilocal villages, while women inherit dominance from their mothers with probability $1-p$ in matrilocal villages. There may be too many dominant individuals after reproduction, especially if $\sigma$ is high. We therefore randomly select $\delta N$ individuals from the set of dominant offspring to form the dominant class in the next generation, with other dominant offspring assigned to the non-dominant class. If there are fewer than $\delta N$ dominant offspring, we repeat the simulation of the reproduction process. This is a computational convenience that will tend to slightly bias upward the average advantage of dominance in a generation when $\sigma$ is small.

Figure 3 retains the sex-biased migration from the earlier analysis, but adds various degrees of social dominance. We apply a fixed proportion of dominant individuals $\delta = 0.06$, while varying migration rates $m$, the selective advantage of dominance $\sigma$ and the heritability of reproduction $1-p$. The proportion of dominant individuals $\delta = 0.06$ is based on levels of dominance previously detected ($0.02 < \delta < 0.06$) in a number of villages showing evidence of dominance [52].

The inclusion of a strong dominance effect is theoretically able to generate villages with the lower and more variable haplotype diversities observed in the genetic data. However, the degree of social

dominance required to approach observations (e.g. figure 3d,e) require high heritability $(1 - p = 0.9)$ and high reproductive advantages $(\sigma = 0.8)$ at rates that are not anthropologically plausible. Despite extensive ethnographies from the region (e.g. [54]), nothing resembling this level of social dominance has been reported.

Furthermore, the observed symmetric pattern of haplotype diversity would require equally strong heritable female dominance in the system of matrilocal villages. While female social dominance certainly exists in some societies, including Wehali [55], its correspondence with reproductive skew is unclear. There are no reports supporting such a strong female reproductive skew in any traditional human communities to our knowledge. We, therefore, conclude that yet other processes are at play, and we proceed with a qualitative fitting of models incorporating both dominance and founder effects.

## 3.4. Qualitative fitting of sociocultural processes

To tease apart the relative impact of sex-biased migration, dominance and founder-ideology effects, we proceed by assessing the fit of simulations to the observed data using models with and without each process. Our implementation of sex-biased migration and social dominance has already been described. To incorporate kin-structured founders, we apply a founder-focused colonization model in which villages grow and attempt to bud upon reaching a specified population size, until there is no more space for further villages (i.e. the number of villages reaches $V$). At this point, villages would cease to bud and, if the simulation were to continue, the system would ultimately relax into the island model.

A village of size $n_i$ undergoes deterministic logistic growth by choosing $n_i(1 + r(1 - n_i/N))$ parents to each contribute a child in the next generation, with parents chosen independently and with replacement with probability $(1 + \sigma)/N(1 + \sigma\delta)$ if dominant and $1/N(1 + \sigma\delta)$ if not dominant. Upon reaching a population size $N_{\text{bud}} < N$, $M$ randomly chosen individuals leave to form a new village. These colonies continue to grow and bud, until there are $V$ villages and the simulation is stopped. Standard random mating or the social dominance model of reproduction are applied when simulating marriage within each subgroup. Mutation and sex-biased migration between villages are included as before.

Our approach is to attempt a qualitative fit of the microevolutionary processes involved in generating diversity. We therefore sought to explore parameters that are anthropologically plausible rather than identifying quantitatively best-fitting sets of many parameters. This vastly reduces the otherwise considerable model space, and offers an example of how qualitative model fitting can be used as a logical tool.

Sex-biased migration rates were retrieved from previous model fitting on similar data, and coupled with the observed number and sizes of villages [38]. We note that the founder simulation runs for around 60 generations before terminating, which corresponds well with genetic estimates of Austronesian influx into the region and subsequent establishment [2,32]. Both the founder model and dominance effect parameters were chosen such that parameter values are consistent with observations reported in the anthropological literature. Model parameters are shown in table 1.

The haplotype diversity implied by the island and founder models with sex-biased migration, and with or without social dominance, is shown in figure 4. The island model, with or without dominance, fails to generate the observed variation in village haplotype diversity, with most simulated villages also having high haplotype diversity. Conversely, the founder model generates simulated villages with a broader range of haplotype diversities, with some probability of creating the extremely homozygous villages observed on the islands. Greater homozygosity tends to be observed for younger villages that have been subject to more iterations of the repeated founder process.

We applied the two-dimensional KS statistic to assess the fit of observed haplotype diversity to each of the four structurally different models. The best fit for patrilocal Sumba was the founder model with dominance ($KS_{\text{Sumba,HD}} = 0.47$), while the best fit for matrilocal Timor was the founder model without dominance ($KS_{\text{Timor,HD}} = 0.30$).

The possibility that social dominance plays an important role in patterns of Y chromosome genetic diversity on Sumba appears to differ from previous results that suggested dominance rarely skews the haplotype frequency spectrum in the region [52]. To assess the validity of our interpretation and the relationship between current and previous results, we therefore additionally calculated Slatkin's test on each village separately. We plot these results against the P values generated using simulated villages for the various models (figure 5). The probability value of Slatkin's exact test ($P_E$) sums the

**Table 1.** Variables for the island model and the founder model. Variables in the first section are common for all four cases in figure 4. Variables in the second section apply only to the island model and founder model when a dominance effect is added. Variables in the third section pertain only to the founder model.

| variable | symbol | value |
|---|---|---|
| number of villages | $V$ | 50 |
| population size | $N$ | 300 |
| sample size | $n_s$ | 24–69 |
| mutation rate – mtDNA | $\mu_{mt}$ | 0.0186 |
| mutation rate – Y-STR | $\mu_Y$ | 0.0249 |
| migration rate – male ♂, matrilocal | $m_{mm}$ | 13.5% |
| migration rate – male ♂, patrilocal | $m_{mp}$ | 0.5% |
| migration rate – female ♀, matrilocal | $m_{fm}$ | 4.5% |
| migration rate – female ♀, patrilocal | $m_{fp}$ | 5.5% |
| fraction of dominant individuals | $\delta$ | 0.06 |
| selective advantage | $\sigma$ | 0.8 |
| heritability of reproduction | $1-p$ | 0.8 |
| population growth rate | $r$ | 0.2 |
| budding population size | $N_{bud}$ | 250 |
| founder population size | $M$ | 50 |

probabilities of the configurations (with fixed sample size and fixed haplotype number) that are less likely than the configuration of the observed haplotype frequency distribution.

As previously reported, few villages reach a significance threshold of $P_E > 0.975$, with only 4 of 14 villages on Sumba showing a statistically significant signal of non-neutrality consistent with social dominance. This supports the earlier assertion that dominance does not lead to universally extreme skews in the Y chromosome haplotype frequency spectrum [52]. However, the overall distribution of P values is skewed. The simulations offer some insight into the nature of this skew, and highlight that Slatkin's test is a useful method to differentiate between different sociocultural processes.

As expected, the island model does not show extreme Slatkin's $P$ value skews, although there is a slightly higher density at high $P_E$ in the Y chromosome distribution in figure 5a. This is due to the characteristic treatment of samples in which all haplotypes are unique (configuration $\mathbf{b} = [n_s, 0, 0, 0, \ldots]$), which are assigned an exact test value of $P_E = 1.0$ despite not indicating an overly uneven distribution. This emphasizes the importance of comparing real and simulated data when drawing inferences from models. Adding dominance generates the expected Slatkin's P value skews. Figure 5b,f show, respectively, the predicted impact of female and male dominance given an island model, leading to correspondingly high mtDNA and Y chromosome $P_E$.

The founder model (figure 5c,d and g,f) tends to generate a weak skew toward higher $P_E$ values for both mtDNA and the Y chromosome, as the serial founder effect impacts both sex-linked loci. Interestingly, there is a slight tendency for the founder model to generate greater skews in Y chromosome diversity in matrilocal Timor and mtDNA diversity in patrilocal Sumba. This is probably due to the greater effective growth rate for the dispersing sex, which ultimately achieves a larger effective population size due to higher migration between villages.

Comparing the observed data to the model predictions (figure 6) reveals strong support for weak dominance on Sumba, but limited resolution to distinguish between models on Timor. As before, identifying the model with the lowest summed KS distances over both haplotype diversity and Slatkin's test supports the founder model with dominance on Sumba ($KS^{FM,D}_{Sumba,HD} + KS^{FM,D}_{Sumba,Slatkin} = 1.05$) and the founder model without dominance on Timor ($KS^{FM,ND}_{Timor,HD} + KS^{FM,ND}_{Timor,Slatkin} = 0.81$). The difference between the non-dominance (0.30 and 0.51) and dominance (0.32 and 0.53) founder models for Timor is not significant. Thus, we do not rule out matrilineal dominance on Timor—the inclusion of dominance fits the data reasonably well—but the observed diversity patterns can be generated without including dominance in the model.

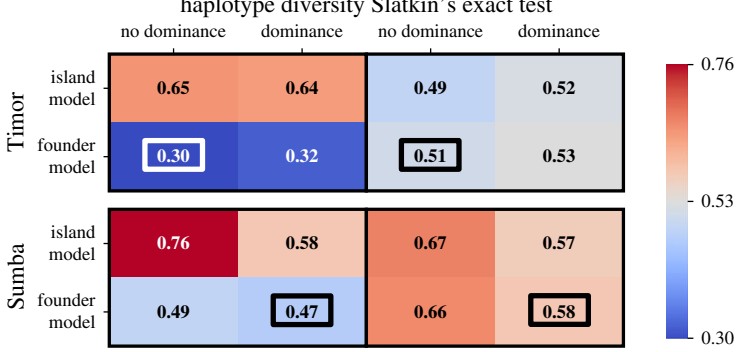

**Figure 6.** Kolmogorov–Smirnov (KS) distances between the data and the four models studied, with parameters as indicated in table 1 and specifications given in the text. The model with the lowest summed KS distance of the two summary statistics studied is boxed.

# 4. Discussion

## 4.1. Fine-scale sampling reveals socio-genetic change on the fly

While the behaviours of individuals can ultimately have a genetic impact at the scale of countries and continents, decisions about marriage, dispersal patterns and the competition for social dominance occur in the context of kinship relations at the scale of villages. This work shows that sampling and analysing data at the fine geographical scale of villages enables a better understanding of the connection between genetic variation and broader anthropological hypotheses (figure 7).

A simple thought experiment illustrates the importance of understanding scale in sampling and analysis when studying sociocultural processes. Consider the implications of re-analysing our samples, with villages pooled to build representative samples of islands (Sumba, Timor) or a regional population (e.g. across both Sumba and Timor). Recalculating the two summary statistics analysed above yields striking differences (table 2). At the regional scale, both mtDNA and the Y chromosome are highly diverse (≥0.97), with a tendency towards lower mtDNA haplotype diversity. The Slatkin test indicates an exceptionally strong skew in haplotype frequency spectra, consistent with both matrilineal (mtDNA) and patrilineal (Y chromosome) dominance. At the scale of individual islands, the haplotype frequency spectra remain consistent with matrilineal and patrilineal dominance, but we additionally observe the lower mtDNA haplotype diversity expected on Timor (due to matrilocality and hence excess mtDNA drift). However, we see no particular deficit in Y chromosome haplotype diversity on Sumba despite evidence for stringent patrilocal marriage rules there [38].

The strikingly different patterns observed when pooling indicate that our ability to determine the relative contributions of different sociocultural factors is heavily scale dependent, with results and interpretations shifting when sampling occurs at regional, island or village levels. If the sampling scheme is not explicitly accounted for, inference based on samples taken at larger scales may be compatible with multiple different combinations of sociocultural processes acting at smaller scales—potentially including false matches to processes that are not actually acting.

## 4.2. Qualitative fitting circumvents model complexity

Using finer sample groupings to better represent interacting sociocultural units leads to special challenges for demographic inference. Formal modelling of individual population sizes, asymmetric migration rates, and the order and timing of village budding events would lead to a complex many-parameter model that is computationally unfeasible to fit, especially when flexible-but-slow forward-time simulation is required. Furthermore, there is the potential that individual parameters with limited anthropological relevance or interest may be emphasized, reflecting historical accident in individual communities rather than fundamental underlying sociocultural processes. In our study, two modelling decisions prove particularly helpful to counteract these limitations.

First, we purposely chose to ignore the details of the relationships between villages. Relationships between individual populations and samples is a standard subject in much of genetic demography. However, focusing instead on the shared microevolutionary processes that are consistent with patterns

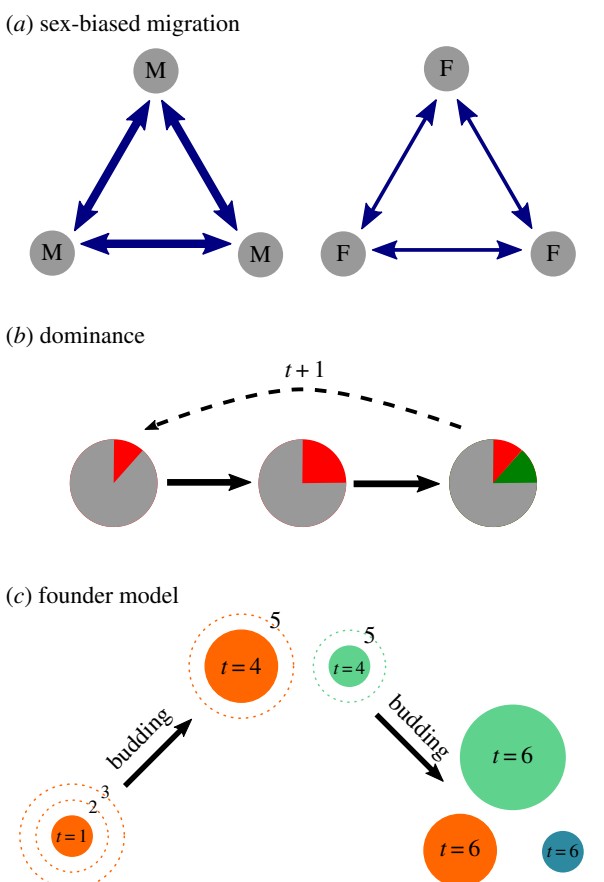

*(a)* sex-biased migration

*(b)* dominance

*t + 1*

*(c)* founder model

**Figure 7.** Schematic of processes included in the sociocultural models. (*a*) Sex-biased migration. In patrilocal societies (shown), men migrate between villages more often than women, while the converse is true for matrilocal societies. (*b*) Dominance. Villages contain a proportion of dominant individuals (red). As dominant individuals reproduce faster than non-dominant individuals, the proportion grows; however, the rules of inheritance mean that dominance is not fully heritable and some dominant individuals in one generation move to the non-dominant class in the next generation (green). (*c*) Founder model. Villages bud and shrink when they reach a certain size. Six generations are shown here, with budding (in this example) in the third generation after the first village was founded.

**Table 2.** Sample haplotype diversity and Slatkin's test results when samples are analysed at island (Timor, Sumba) or regional (Timor and Sumba) scales. Pooling tends to increase observed haplotype diversity, hide genetic evidence of kinship systems (e.g. Y chromosome haplotype diversity is similar on both matrilocal Timor and patrilocal Sumba), and generate highly skewed haplotype frequency spectra (Slatkin's test).

| | samples | | number of haplotypes | | sample haplotype diversity | | Slatkin's $P_E$ | |
|---|---|---|---|---|---|---|---|---|
| | mtDNA | Y | mtDNA | Y | mtDNA | Y | mtDNA | Y |
| Sumba | 634 | 646 | 162 | 209 | 0.980 | 0.978 | 0.999 | 0.999 |
| Timor | 450 | 421 | 111 | 235 | 0.948 | 0.991 | 0.999 | 0.999 |
| Sumba and Timor | 1084 | 1067 | 243 | 431 | 0.977 | 0.990 | 1.000 | 1.000 |

of observed diversity enabled us to gain more pertinent insight into human behaviour. While sociocultural processes are not independent of demography—indeed, our simulations of founder effects follow a non-equilibrium model—studying islands with contrasting kinship systems but similar genetic histories lets us focus on important cultural behaviours that vary between the communities.

Second, we take a modelling approach where properties of the model are iteratively switched on and off, rather than keeping the model structure fixed and seeking to closely fit model parameters. By

choosing parameter values that are both (i) anthropologically plausible and (ii) sufficiently strong to represent behaviourally meaningful change, we can rapidly explore alternative sociocultural mechanisms, using simulation as a thinking tool rather than over-fitting models that inevitably represent an incomplete picture of reality.

Together, these approaches enable us to circumvent computational complexity, while still gaining novel insight into the sociocultural processes of the greater Pacific region.

## 4.3. Sociocultural behaviours impact variation across scales

Our approach allowed us to explore the impact of progressively more complex combinations of sociocultural processes on genetic variation, and to assess which of these models correspond well to diversity patterns in 23 villages from two Indonesian island communities that follow different marriage rules.

We found strong support for a cultural complex of interacting social processes—sex-biased migration, serial lineage-based founder effects, and social dominance in patrilocal Sumba but not necessarily in the matrilocal region of Timor—that correspond closely to behaviours predicted by Bellwood's 'founder ideology'. Interestingly, we see this effect within individual islands, suggesting that cultural behaviours similar to those thought to drive long-distance movements between islands play a role in the spread of populations and the interactions of villages within islands too. Decisions—marriage dispersal and village budding, but also movements between islands—are inevitably made within a cultural context, thus generating the sociocultural behaviours that we have modelled. These behaviours have important effects at the smallest scale of genetic variation, in villages, but also lead to the large-scale patterns studied in archaeology and anthropology, such as here in the context of the Austronesian expansion.

Ethics. Genetic samples were obtained by S.G.M., J.S.L. and H.S. and by Wuryantari Setiadi, Loa Helena Suryadi and Meryanne Tumonggor of the Eijkman Institute, with the assistance of Indonesian Public Health clinic staff, following protocols for the protection of human subjects established by Institutional Review Boards at the Eijkman Institute (EIMB no. 90), Nanyang Technological University (2014-12-011) and the University of Arizona (07-0441). Written informed consent was obtained from all participants. Permission to conduct research in Indonesia was granted by the Indonesian Institute of Sciences and by the Ministry for Research, Technology and Higher Education.

Data accessibility. All data are publicly available. Mitochondrial DNA sequences can be downloaded from NCBI GenBank (accession nos. KC113641–KC115854, KJ936094–KJ936619). Y chromosome haplotypes are available in the electronic supplementary material datasets of Tumonggor et al. [21,22].

Authors' contributions. J.S.L. and M.P.C. planned the study; N.N.C. and G.S.J. performed analyses; J.S.L., S.G.M., L.Y.C. and M.P.C. interpreted analyses; N.N.C., G.S.J., J.S.L. and M.P.C. wrote the manuscript with contributions from the other authors. All authors gave final approval for publication.

Competing interests. We declare we have no competing interests.

Funding. Financial support came from a National Science Foundation grant no. SES 0725470 to J.S.L., a Singapore Ministry of Education Tier II grant no. MOE2015-T2-1-127 to J.S.L. and M.P.C., and a Royal Society of New Zealand Marsden grant no. 17-MAU-040 to M.P.C.

Acknowledgements. M.P.C. was supported by a fellowship from the Alexander von Humboldt Foundation, Germany and G.S.J. was supported by an NTU Presidential Postdoctoral Fellowship.

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
