## [Reviewer comments · Royal Society Open Science]

Review History

RSOS-190733.R0 (Original submission)

Review form: Reviewer 1

Is the manuscript scientifically sound in its present form?

Yes

Are the interpretations and conclusions justified by the results?

Yes

Is the language acceptable?

Yes

Do you have any ethical concerns with this paper?

No

Have you any concerns about statistical analyses in this paper?

No

Recommendation?

Accept with minor revision (please list in comments)

Comments to the Author(s)

The authors present data of genetic variation which allow to test different models of socio-genetic interactions involving sex-biased migration, lineage-focused founder effects, and heritable social dominance. The subject is interesting and the data sound.

Here are a few minor comments:

- a) I always find weird that population geneticists keep using 'heterozygosity' for haploid loci.
- b) the color coding in fig 2A (ie from 0 to 600 for matrilocal is not a density and the number should be explain in the figure legend, this applies to subsequent figures
- c) p7 delta should be explained line 48 in addition to table 1. An explanation of the value chosen (0.06) is welcomed, a reference would be even better.
- d) First ocurrence of PE page 10, it would be nice to define.
- e) Fig 6: I would like a discussion about the significance of the difference of 0.3 and 0.32 in the Timor founder model (and 0.51 and 0.53 in Slatkin's exact test)

Review form: Reviewer 2

Is the manuscript scientifically sound in its present form?

Yes

Are the interpretations and conclusions justified by the results?

Yes

Is the language acceptable?

Yes

Do you have any ethical concerns with this paper?

No

Have you any concerns about statistical analyses in this paper?

No

Recommendation?

Accept with minor revision (please list in comments)

Comments to the Author(s)

To the Authors:

In "Sex-linked genetic diversity originates from persistent sociocultural processes at microgeographic scales" Chung, Jacobs and colleagues analyse mtDNA (540bp) and Y chromosome (12 STRs + haplotype informative SNPs) for ~1,100 individuals from two Indonesian islands characterised by opposite societal systems.

In doing so, they observe a marked opposite skewness in the observed heterozygosity (H) in matrilocal and patrilocal populations and used different simulation models to identify the best model explaining the observed heterozygosity. Testing the combination of i) post-marital movements, ii) social dominance and iii) founder-ideology patterns the authors found that a founder model with dominance can approximate the observed H pattern in the patrilocal Sumba,

while the same model with No dominance provides a better fit for the matrilineal Timor, although there is a limited power for the latter population.

The work is well conceived and written, and the methods are rigorous, although some clarifications and additional figures summarising the simulated model could help to raise the interest of the anthropologists' community. Overall, I would recommend the publication of the manuscript after some modifications (see comments to the author).

Here are more specific comments:

P2. Introduction: The introductory paragraph is too concise, and do not adequately summarise what is the actual state of the art in using genetics to characterise the impact of social strategies. Please consider to expand it.

P2, L8 : This sentence seems truncated, please consider to rephrase it.

P2, L12-14: The few works mentioned in this sentence should be cited.

P2, L50. The term Heterozygosity usually refers to an individual rather than a population metric. Given that the work is expected to be of interest to the anthropologists' community, I would consider adding a brief explanation. For the same reason, it would be very useful to include a figure summarising the simulated models.

Decision letter (RSOS-190733.R0)

17-Jul-2019

Dear Dr Cox

On behalf of the Editors, I am pleased to inform you that your Manuscript RSOS-190733 entitled "Sex-linked genetic diversity originates from persistent sociocultural processes at microgeographic scales" has been accepted for publication in Royal Society Open Science subject to minor revision in accordance with the referee suggestions. Please find the referees' comments at the end of this email.

Both reviewers are very positive about publication of the manuscript but they also suggest some minor revisions to your manuscript. Therefore, I invite you to respond to the comments and revise your manuscript.

- Ethics statement

- Data accessibility

It is a condition of publication that all supporting data are made available either as supplementary information or preferably in a suitable permanent repository. The data accessibility section should state where the article's supporting data can be accessed. This section should also include details, where possible of where to access other relevant research materials

such as statistical tools, protocols, software etc can be accessed. If the data has been deposited in an external repository this section should list the database, accession number and link to the DOI for all data from the article that has been made publicly available. Data sets that have been deposited in an external repository and have a DOI should also be appropriately cited in the manuscript and included in the reference list.

If you wish to submit your supporting data or code to Dryad (<http://datadryad.org/>), or modify your current submission to dryad, please use the following link:
<http://datadryad.org/submit?journalID=RSOS&manu=RSOS-190733>

- **Competing interests**

- **Authors' contributions**

- **Acknowledgements**

- **Funding statement**

Because the schedule for publication is very tight, it is a condition of publication that you submit the revised version of your manuscript before 26-Jul-2019. Please note that the revision deadline will expire at 00.00am on this date. If you do not think you will be able to meet this date please let me know immediately.

To revise your manuscript, log into <https://mc.manuscriptcentral.com/rsos> and enter your Author Centre, where you will find your manuscript title listed under "Manuscripts with Decisions". Under "Actions," click on "Create a Revision." You will be unable to make your

revisions on the originally submitted version of the manuscript. Instead, revise your manuscript and upload a new version through your Author Centre.

Once again, thank you for submitting your manuscript to Royal Society Open Science and I look

forward to receiving your revision. If you have any questions at all, please do not hesitate to get in touch.

Kind regards,

on behalf of Dr Alecia Carter (Associate Editor) and Steve Brown (Subject Editor)
 openscience@royalsociety.org

Reviewer comments to Author:

Reviewer: 1

Comments to the Author(s)

The authors present data of genetic variation which allow to test different models of socio-genetic interactions involving sex-biased migration, lineage-focused founder effects, and heritable social dominance. The subject is interesting and the data sound.

Here are a few minor comments:

- a) I always find weird that population geneticists keep using 'heterozygosity' for haploid loci.
- b) the color coding in fig 2A (ie from 0 to 600 for matrilocal is not a density and the number should be explain in the figure legend, this applies to subsequent figures
- c) p7 delta should be explained line 48 in addition to table 1. An explanation of the value chosen (0.06) is welcomed, a reference would be even better.
- d) First ocurrence of PE page 10, it would be nice to define.
- e) Fig 6: I would like a discussion about the significance of the difference of 0.3 and 0.32 in the Timor founder model (and 0.51 and 0.53 in Slatkin's exact test)

Reviewer: 2

Comments to the Author(s)

To the Authors:

In “Sex-linked genetic diversity originates from persistent sociocultural processes at microgeographic scales” Chung, Jacobs and colleagues analyse mtDNA (540bp) and Y chromosome (12 STRs + haplotype informative SNPs) for ~1,100 individuals from two Indonesian islands characterised by opposite societal systems.

In doing so, they observe a marked opposite skewness in the observed heterozygosity (H) in matrilocal and patrilocal populations and used different simulation models to identify the best model explaining the observed heterozygosity. Testing the combination of i) post-marital movements, ii) social dominance and iii) founder-ideology patterns the authors found that a founder model with dominance can approximate the observed H pattern in the patrilocal Sumba, while the same model with No dominance provides a better fit for the matrilocal Timor, although there is a limited power for the latter population.

The work is well conceived and written, and the methods are rigorous, although some clarifications and additional figures summarising the simulated model could help to raise the

interest of the anthropologists' community. Overall, I would recommend the publication of the manuscript after some modifications (see comments to the author).

Here are more specific comments:

P2. Introduction: The introductory paragraph is too concise, and do not adequately summarise what is the actual state of the art in using genetics to characterise the impact of social strategies. Please consider to expand it.

P2, L8 : This sentence seems truncated, please consider to rephrase it.

P2, L12-14: The few works mentioned in this sentence should be cited.

P2, L50. The term Heterozygosity usually refers to an individual rather than a population metric. Given that the work is expected to be of interest to the anthropologists' community, I would consider adding a brief explanation. For the same reason, it would be very useful to include a figure summarising the simulated models.

Author's Response to Decision Letter for (RSOS-190733.R0)

See Appendix A.

Decision letter (RSOS-190733.R1)

26-Jul-2019

Dear Dr Cox,

I am pleased to inform you that your manuscript entitled "Sex-linked genetic diversity originates from persistent sociocultural processes at microgeographic scales" is now accepted for publication in Royal Society Open Science.

Kind regards,

on behalf of Dr Alecia Carter (Associate Editor) and Steve Brown (Subject Editor)
openscience@royalsociety.org

Appendix A

Reply to Reviewers

We thank the editor and reviewers for reading our manuscript, and for their careful and thoughtful comments. We have made a number of changes to the manuscript to directly address the points they raise. These focus on two main areas. First, general improvements in readability, emphasizing points specifically noted by the reviewers. And second, clarifying questions about the form of the models. We believe that these changes address the points that the reviewers raised, and we are grateful for their help in making this work stronger.

Reviewer 1

The authors present data of genetic variation which allow to test different models of socio-genetic interactions involving sex-biased migration, lineage-focused founder effects, and heritable social dominance. The subject is interesting and the data sound.

Here are a few minor comments:

a) I always find weird that population geneticists keep using 'heterozygosity' for haploid loci.

Although commonly used, we agree that 'heterozygosity' can be a confusing term in the context of haploid loci, especially for readers outside population genetics. To make the text more accessible, we now use the term 'haplotype diversity' in the text and figures instead.

b) the color coding in fig 2A (ie from 0 to 600 for matrilocal is not a density and the number should be explain in the figure legend, this applies to subsequent figures.

This is a good point. We now clarify in the figure legends what the contours mean. In short, contour lines are estimated based on 10^4 data points simulated under the relevant model. The data points are distributed within a 100 x 100 uniform grid. The probability value for the contour line at a given location (x, y) , after being divided by the simulation size (10^4), thus gives the probability of a data point being found within the grid $(x-0.0025, y-0.0025, x+0.0025, y+0.0025)$.

c) p7 delta should be explained line 48 in addition to table 1. An explanation of the value chosen (0.06) is welcomed, a reference would be even better.

The meaning of δ – the fraction of the population that has a selective advantage over the nondominant population – is now given in the text, in addition to Table 1. We also provide a reference explaining the simulated value of 0.06.

d) First occurrence of PE page 10, it would be nice to define.

Yes! We thank the reviewer for noting this omission. The variable P_E is now defined at first use in the text.

e) Fig 6: I would like a discussion about the significance of the difference of 0.3 and 0.32 in the Timor founder model (and 0.51 and 0.53 in Slatkin's exact test)

We agree. We now explicitly discuss these differences (0.3 vs 0.32, and 0.51 vs 0.53), and emphasize that they are not significant. Note that this was our original interpretation; we are simply now making this point clearer in the text.

Reviewer 2

In “Sex-linked genetic diversity originates from persistent sociocultural processes at microgeographic scales” Chung, Jacobs and colleagues analyse mtDNA (540bp) and Y chromosome (12 STRs + haplotype informative SNPs) for ~1,100 individuals from two Indonesian islands characterised by opposite societal systems.

In doing so, they observe a marked opposite skewness in the observed heterozygosity (H) in matrilocal and patrilocal populations and used different simulation models to identify the best model explaining the observed heterozygosity. Testing the combination of i) post-marital movements, ii) social dominance and iii) founder-ideology patterns the authors found that a founder model with dominance can approximate the observed H pattern in the patrilocal Sumba, while the same model with No dominance provides a better fit for the matrilocal Timor, although there is a limited power for the latter population.

The work is well conceived and written, and the methods are rigorous, although some clarifications and additional figures summarising the simulated model could help to raise the interest of the anthropologists' community. Overall, I would recommend the publication of the manuscript after some modifications (see comments to the author).

Here are more specific comments:

P2. Introduction: The introductory paragraph is too concise, and do not adequately summarise what is the actual state of the art in using genetics to characterise the impact of social strategies. Please consider to expand it.

This is a good point. We have now expanded the Introduction to provide a lengthier summary of how genetics can be used to characterize social strategies. We have also added several new citations, which cover a broader range of publications across this subject area.

P2, L8 : This sentence seems truncated, please consider to rephrase it.

We have rephrased this sentence.

P2, L12-14: The few works mentioned in this sentence should be cited.

We have cited these publications. Note that this sentence differs slightly from the original version as part of the wider changes made to the Introduction as requested above.

P2, L50. The term Heterozygosity usually refers to an individual rather than a population metric. Given that the work is expected to be of interest to the anthropologists' community, I would consider adding a brief explanation. For the same reason, it would be very useful to include a figure summarising the simulated models.

As noted above, the term ‘heterozygosity’ has been replaced by ‘haplotype diversity’ for improved readability.

We have also included a new figure (Figure 7), which gives a more general graphical overview of the simulated models, particularly for an anthropological audience.